# Optimizing costs and sustainability for gamma knife radiosurgery: A cost and breakeven analysis at India's largest neurosurgery centre

Nitin Agrawal[1], Nishant Sharma[2], Tilotma Jamwal[2], Kshitija Singh[2], Vijaydeep Siddharth[3]*, Sidhartha Satpathy[2]

1 Max Super-Speciality Hospital, Shalimar Bagh, Delhi, India, 2 Department of Hospital Administration, All India Institute of Medical Sciences (AIIMS), New Delhi, India, 3 Department of Hospital Administration & Additional Medical Superintendent, JPNA Trauma Centre, AIIMS, New Delhi, India

* vijaydeep@aiims.edu

**Data Availability Statement:** All relevant data are within the manuscript and its Supporting Information files.

## Abstract

### Background

Establishing and maintaining gamma knife facility incurs significant costs, mandating healthcare institutions to meticulously assess financial implications for sustainability.

### Methods

This study explores the financial implications of setting up and operating a Gamma Knife facility, with an aim to ascertain user charges for achieving breakeven. The study was conducted from January to June 2019 at the largest neurosurgery centre of an Institute of National Importance (INI), in Delhi, India. Applying both Traditional and Time-Driven Activity-Based Costing (TD-ABC) methodologies, capital and operating expenses were calculated. (1US$ = INR 70.4 –Average value for the year 2019).

### Results

The average cost per gamma knife radio surgery procedure was calculated to be US $2,469 (INR 1,73,832), with major costs attributed to machinery & equipment (43.6%), followed by manpower (32.5%), electricity (9.67%), equipment maintenance (8.61%) etc. The initial investment to establish a Gamma Knife facility is significantly higher with an MRI unit (Model A) at US $ 9,836,423 (INR 69,24,84,164) compared to one without (Model B) at US $7,294,986 (INR 51,35,66,988). Currently, the patient has to pay US $1,065 (INR 75,000) for a gamma knife radio surgery, which poses a challenge for achieving breakeven since the variable cost for the same is US $1,367 (INR 96,239) per procedure.

### Conclusion

The study serve as a tool for strategic planning, pricing adjustments, and enhancing operational efficiencies, thus ensuring that such high-end technologies can be sustainably integrated into the public healthcare landscape of a developing country like India.

**Funding:** The author(s) received no specific funding for this work.

**Competing interests:** The authors have declared that no competing interests exist.

## Introduction

Healthcare Institutes often need to invest in advanced equipment and facilities to provide high-quality medical services and stay at the forefront of technological advancements. The gamma knife radio surgery (GKRS), utilizes state-of-the-art technology involving radiation treatment for treating small lesions with pinpoint accuracy using three-dimensional "stereotactic" imaging while delivering multiple thin radiation beams through an arc or sphere targeting tumour as the focal point causing minimum damage to the tissues surrounding the tumour, for precise and non-invasive treatment of various neurological conditions [1, 2]. Research on Stereotactic Radiotherapy (SRT) shows that more than 20% of patients encounter severe financial difficulties shortly after initiating the treatment, indicating a high incidence of financial toxicity (Profound financial distress experienced by individuals undergoing cancer therapy) [3]. On the other hand, establishing and maintaining gamma knife facility incurs significant costs, mandating healthcare institutions to meticulously assess financial implications for sustainability. Hospital operating costs are a major factor contributing to the widening financial gap in the healthcare sectors of many countries [4]. In countries with developing economies, such as India, hospitals are regarded as an essential community asset and has a social obligation of delivering needed healthcare services to the community which are expected to be of satisfactory quality and provided at the lowest feasible cost. The process of cost analysis assists Hospital Administrators/Managers, and policymakers in evaluating the effectiveness of their facilities in fulfilling these obligations [5].

The methodologies of cost analysis are instrumental for hospitals to distribute direct and indirect costs accurately, thereby deriving the actual cost of services. Understanding cost analysis aids in budget preparation and in deciding user charges that reflect real costs is a necessity for a hospital's financial health. It serve as a foundational tool for ensuring that operational expenses align with available funds and subsidies, thereby underscoring their significance in sustainable healthcare management [6]. Strategic decisions are crucial for the sustained availability and success of advanced medical technologies. Considering the significance of breakeven analysis in determining the financial sustainability of the gamma knife facility, it is essential to delve into the specific cost components associated with its establishment and operation. While traditional costing methods may provide a general overview of overhead allocation, the implementation of time driven activity based costing (TDABC) will offer a more precise understanding of resource consumption for each activity involved in the facility's operation. Availability of the data determines the kind of method to be deployed for cost analysis. Utilizing these costing methods, it becomes possible to deduce the user charges required to achieve breakeven analysis and ensure the financial sustainability [7]. GKRS was started in May 1997 in the Neurosurgery Centre of All India Institute of Medical Sciences (AIIMS), New Delhi, India, first in the region in a public sector hospital. As already known Gamma knife facility is a cost-intensive setup, this study was carried out with an aim of identifying various cost centres for estimating the cost of establishing and maintaining a gamma knife facility with a focus on determining the user charges required to achieve breakeven analysis.

## Materials & methods

An exploratory and observational study was conducted in the Gamma Knife Facility of Neurosciences Centre at AIIMS (Institute of National Importance) in New Delhi, India from Jan to June 2019 which included process analysis, interaction with concerned stakeholders and retrospective record review. Combination of both traditional and time driven activity based costing methods were used for cost analysis. The primary data on resource utilisation and identifying various cost centres was collected through unstructured interview with healthcare workers

**Table 1. Cost centres included and excluded in the study for cost analysis of GKRS.**

| Cost centres included in the study | |
|---|---|
| **Capital Cost/ Fixed Cost\*** | **Operating Cost/ Variable Cost#** |
| Building | Building maintenance$ |
| Machinery & Equipment | Equipment maintenance$ |
| Furniture and fixtures | Human resource |
| | Consumables (Medical, surgical, general etc.) |
| | Heating Ventilation and Air Conditioning (HVAC) |
| | Electricity |
| | Support Services |
| **Cost centres excluded in the study** | |
| Land (Provided free of cost by the Government of India) | |
| Administrative Cost | |
| Computer Facility | |
| Medical Records | |
| IT Cost | |
| Water supply | |
| Medical Records | |

\*Fixed costs are the expenses that do not vary with the quantum of service provided (for e.g. land, machinery & equipment etc.) [9].

#Variable costs are the expenses which directly vary with the amount of service provided, or number of procedures performed (for e.g consumables, human resource etc.) [9].

$ Building and equipment maintenance cost are primarily fixed but may exhibit some variability over a period of time due to unforeseen repairs or upgrades.

Replacement cost of various items were arrived by bringing the historical costs to the study year i.e. 2019 by employing Cost Inflation Index (CII) published by Government of India [10]. Being able to categorize costs appropriately is an important step for budgeting and in determining the breakeven point (i.e., where the service/ production cost is equal to the revenue being generated, there onwards service/production starts generating surplus or profit).

(faculty members, resident doctors, medical physicists, support staff etc.) and secondary data on resource utilisation was retrieved from records of various hospital areas (patient care areas, accounts department, engineering services, purchase section etc.). In consonance with the WHO guidance on cost analysis [8] the costs were divided into capital and operating cost (Table 1). All calculations were done using Microsoft Excel 2018 version.

## Capital cost

Total area of the Gamma Knife facility including Magnetic Resonance Imaging (MRI) facility is 310 sq.mt. As the land for the Gamma Knife Facility was provided free of cost by the Government of India, no cost was attributed to it. Construction cost was arrived using replacement method of costing based on the Central Public Works Department (CPWD) rates for building construction [11]. The life of the building as per the CPWD maintenance manual was 75 years and thus the building was depreciated 1.33% annually [12] (Table 2). The procurement of machinery and equipment cost was taken from the purchase department and replacement cost was arrived. Annualized cost was calculated using following steps:

a.  Identification of cost centers

b.  Calculation of replacement cost

**Table 2. Useful life and depreciation method for arriving at annual cost of capital assets.**

| Capital Assets | Life in years | Annualization method |
|---|---|---|
| Building | 75 years | Straight line depreciation |
| Machinery & Equipment | 10 years | Replacement Cost /Annualization factor (8.3) |
| Fixed Machinery/equipment | 15 years | Straight line depreciation |
| Furniture and fixtures | 5 years | Straight line depreciation |

c. Calculation of annualization factor

$= (1/r)^*[1-1/(1+r)^n]$ where r = real interest rate n = no. of life year
Real interest rate (r) = (1+nominal interest rate)/(1+annual inflation)-1

a. Capital cost in year 2018–19 = Replacement cost in year 2018-19/ Annualization factor

Nominal interest rate are the bank lending rates and was taken as 6% in this study, and thereafter real interest was calculated to be 2.9~3%. In general, it is suggested we use a real interest rate of 3%, as has been found in a comprehensive set of cost effectiveness studies for the health sector (both developed and developing countries) and its use makes hospital costing consistent with international literature [8].

The cost of furniture and fixtures were taken from the Government E Marketplace (An online portal for government purchases). Cost of wood work was acquired from the Civil Engineering Department and maintenance cost of 5% was added to furniture, thereafter annual cost was derived using straight line depreciation method.

## Operating cost

As per repair & service cost index issued by office of Director General, Central Public Works Department, Government of India (2018), the maintenance rate was US$ 95 (INR 6,668) per sq. mt. per year for the hospital building. This rate was used to estimate the annual cost that would be incurred in maintaining the Gamma Knife Facility [13]. Equipment Maintenance cost was derived from the purchase agreement available with the hospital.

Staff directly and indirectly committed and contributing to the functioning of Gamma Knife facility included doctors (Faculty and Residents), Nurses, Medical Physicists, Clerk, Hospital Attendants, Sanitary Attendants and Security Staff. Salaries of various cadre of healthcare workers as mentioned above was taken and were apportioned based on the time dedicated towards the functioning of Gamma Knife Facility, which was obtained through direct observation and interaction with healthcare workers.

An exhaustive list of various consumables used for different procedures in the Gamma Knife Facility was obtained and consumption of the same over a period of one year was retrieved from Hospital Information Management System (HIMS). Purchase cost of various consumables was obtained from the hospital stores. Since there were no sub-meter for measuring electricity load, cost of electricity was derived from the electricity consumption of the machines and equipment. Wattage of the equipment and electrical fixtures was taken in the consultation with the engineering department. The number of hours of their usage was taken from the concerned area staff. The total number of unit consumed by all the equipment and fixtures in various areas were calculated and multiplied with the unit cost of electricity.

The cost of manifold gases was taken from a study done by the Department of Hospital Administration within the Institute [14] which was then adjusted using cost inflation index. The number of total manifold points for each area was calculated and cost per manifold point

was taken as US$ 0.36 (INR 25). The cost of laundry services was calculated by taking the list of items that were sent to the laundry in the month of April 2019 and then cost was ascertained by multiplying each item with its rates provided by the laundry. The cost of the biomedical waste management (BMW) was calculated using the Service Level Agreement with the out-sourced vendor.

## Unit cost per GKRS

The total number of Gamma Knife Radiosurgery procedures that were performed over a period of one year (from April 2018-March 2019) was obtained from the HMIS of Gamma Knife Facility and the following formula was used to derive the unit cost:

$$\frac{\textbf{Total annualized capital cost + Total annualized operating cost}}{\textbf{Total procedures done in a year}} = \textbf{Unit cost of Procedure}$$

Calculations were made for establishing the Gamma Knife Facility with or without MRI Facility and breakeven analysis was also carried out assuming various scenarios. Five years cost was used for calculating breakeven analysis. This time frame was selected based on the projection that all the investments would be recouped within this period and thereafter, the facility is expected to start generating surplus considering that the life span of equipment is estimated to be 10 years.

The formula for used for calculating break-even point is as below:

$$\textbf{Break-Even Point} = \frac{\textbf{Fixed Costs}}{\textbf{(Revenue per Unit–Variable Cost per Unit)}}$$

Average US$ value for the year 2019 was taken for the conversion of Indian Rupee (INR) to US $ costs for better appreciation of cost analysis.

## Results

AIIMS located in New Delhi, India is an Institute of National Importance was established in the year 1956 with the trinity of mission of Education, Research and Patient care in that order, is housing the largest Neurosurgery Centre in the country. It has 205 inpatient beds including two intensive care units (ICUs) having 23 beds. It has an annual outpatient of 150,000 (including emergency patients), admits approx. 10000 patients and carries out approx. 4000 neurosurgical procedures every year. The Centre offers world class neurosurgical facilities in all neurosurgical sub-specialties, which includes vascular surgery, epilepsy surgery, minimally invasive cranial and spinal neurosurgery, paediatric neurosurgery, functional neurosurgery, neuro-oncosurgery, skull base & complex spinal surgeries etc. It has advanced imaging modalities and navigational devices (like Stealth Station and ROSA). Neurosurgery healthcare services at the centre are supported by the Department of Neurology (distinct infrastructure), Neuro-Anaesthesia and Neuro-Radiology. It has an average bed occupancy of 80%, 12 days of average length of stay and net death rate of 4.1%.

## GKRS process flow

For the GKRS, patient first reports to the neurosurgery OPD and after detailed evaluation is referred to the Gamma Knife Facility (Fig 1). Patient is provided with an estimated expenditure which is fixed and includes the cost of imaging (MRI/CT/Angiography/Contrast material) and gamma knife procedure. No extra charges are levied if the dose is given in multiple sittings (fractionation). Patient is given an appointment date and is called a day before the procedure for workup. The patient reports with previous scans like MRI, CT, etc. and additional relevant investigations may also be conducted depending upon the diagnosis like Digital Subtraction Angiography for arteriovenous malformation, MRI brain for any neurological tumour, pure

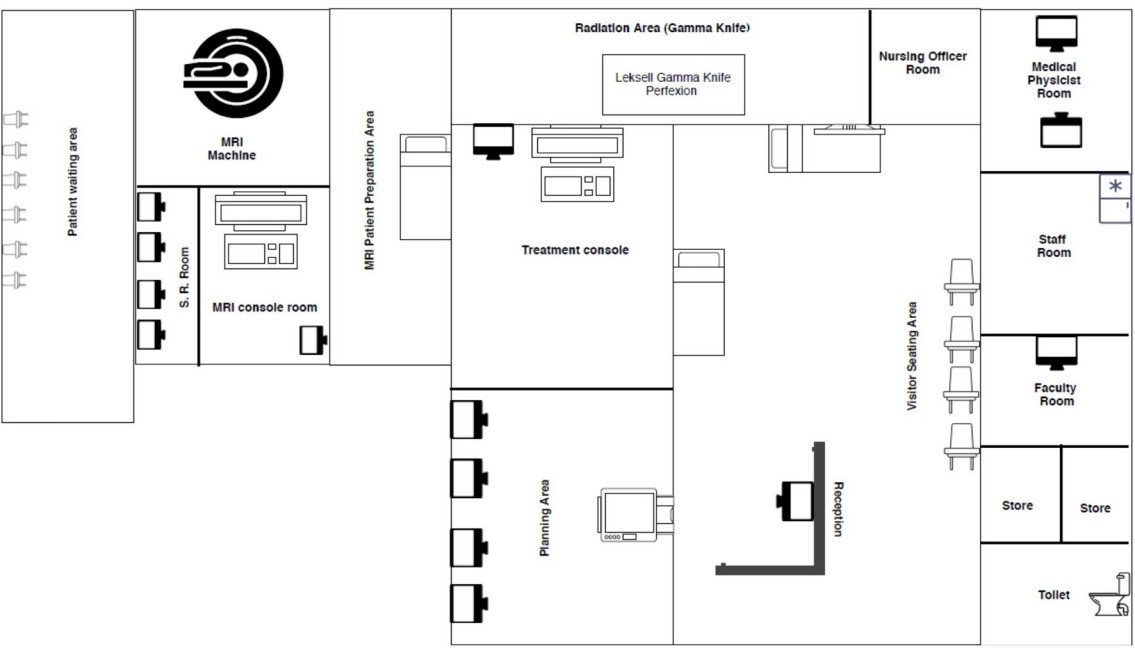

**Fig 1. Representational diagram of gamma knife facility at neurosurgery centre AIIMS, New Delhi.**

tone audiometry for vestibular schwannoma (Acoustic Neuroma) and hormonal assay for pituitary adenoma. The patient presents on the day of the treatment to the Gamma Knife Facility. An aluminium based frame with indicator box is fixed on the patient's head by the neurosurgeon assisted by the Nursing Officer and the Radiation Therapy Technologist. After this the patient is taken to the MRI room where the contrast material is administered and then shifted to the MRI machine. The indicator box on the aluminium frame gives an idea about the stereotactic location of the tumour and a true axial cut is taken with reference to the frame.

MRI Images are transferred through PACS to the planning room where the medical physicist defines the target tissue where the gamma radiation is to be given. Here, the various coordinates are defined for the target as per a three dimensional coordinates system on the Leksell Gamma Plan software to delineate it from the nearby critical organs like optic nerve and chiasm, brainstem etc. The dose calculation is done by a Radiation Oncologist and also the duration of treatment is decided. Once the planning is completed the plan is transferred to the console. The patient is then taken to the Gamma Knife machine where the frame on the patient's head is attached to the adapter on the machine and the patient lays down on the treatment couch of the machine. During the entire process the patient is being monitored from the treatment room. Once the treatment has been completed by the machine the patient is advised a little rest.

## Cost analysis

For calculation of unit cost of gamma knife procedure, it was assumed that the number of procedures being carried out in the facility was taken as the optimal utilisation in the given resources and patient load of the neurosurgery centre. The total annual cost of Gamma Knife Facility was calculated to be US $1,607,453 (INR 113,164,674) for the year 2018–19 (Table 3). Of the total cost, 55% was the operating cost, while remaining was the capital cost (Fig 2). The

**Table 3. Annual cost of gamma knife facility at neurosurgery centre at AIIMS, New Delhi for the year 2018–19.**

| S. No. | Particulars | Total/year | % of cost centre | |
|---|---|---|---|---|
| | **CAPITAL COST** | | | |
| 1 | Building cost | US $1615 (INR 1,13,701) | 0.10 | 45% |
| 2 | Machinery & Equipment<br>Leksell Gamma Knife model Perfexion + Extended System for treating large volume tumors + Gamma Plan workstation+ Dedicated MRI/CT scanner & data storage system capital cost US $554,380 (INR 3,90,28,382) Optima MR 450w with GEM system (Turnkey) US $146,490 (INR 1,03,12,922) | US $ 700,871 (INR4,93,41,304) | 43.60 | |
| 3 | Furniture and fixtures | US $14,171 (INR 9,97,661) | 0.88 | |
| 4 | HVAC cost | US $ 852 (INR 60,000) | 0.05 | |
| | **OPERATING COST** | | | |
| 5 | Building maintenance Cost | US $ 29,364 (INR 20,67,253) | 1.83 | 55% |
| 6 | Equipment maintenance Cost (Leksell Gamma Knife INR 69,15,857.28, 1.5 T MRI machine INR 28,32,000 and TLD badges (10) INR 1000) | US $ 138,478 (INR 97,48,857) | 8.61 | |
| 7 | Human resource Cost | US $ 522,359 (INR 3,67,74,113) | 32.50 | |
| 8 | Consumables Cost | US $ 20,752 (INR14,60,945) | 1.29 | |
| 9 | HVAC maintenance (INR 1008000) and operation cost (INR 360000) | US $ 19,432 (INR13,68,000) | 1.21 | |
| 10 | Electricity Cost (including HVAC) | US $ 155,431 (INR 1,09,42,323) | 9.67 | |
| 11 | Support services Cost | US $ 4127 (INR 2,90,513) | 0.26 | |
| | Total US $ 1,607,452 (INR 11,31,64,670) | | 100 | |

total procedures done during this period were 651, and hence, the unit cost per procedure was calculated to be US $2,469 (INR 1,73,832).

For establishing a Gamma Knife Facility, two types of models were considered: Model A–Gamma Knife Facility having a MRI facility (Table 4) wherein, the total cost of establishing and operating a new Gamma Knife Facility over a five year period amounts to US $ 9,836,423 (INR 69,24,84,164). While, Model B has gamma-knife Facility only and MRI facility is not available in house. The total cost of establishing and operating a new Gamma Knife Facility (without MRI) over a five year period amounts to is US $7,294,986 (INR 51,35,66,988) (Table 5).

## Break even analysis

During the study period, the Gamma Knife facility levied user charges of US $ 1,065 (INR 75,000) per procedure inclusive of repeat visits for the same procedure. In the study it was observed that the variable cost per procedure was US $1,367 (INR 96,239), hence, the Gamma Knife Facility is not bound to attain breakeven, as the charges levied to the patient (US $1,065) are less than the variable cost per procedure (US $1,367). Three scenarios have been described for estimating the break-even points with different charges being levied from the patients and by altering the working hours (Table 6). Here, the fixed cost = US$ 717,509 (INR5,05,12,665) (Capital Cost) is constant for all the three scenarios.

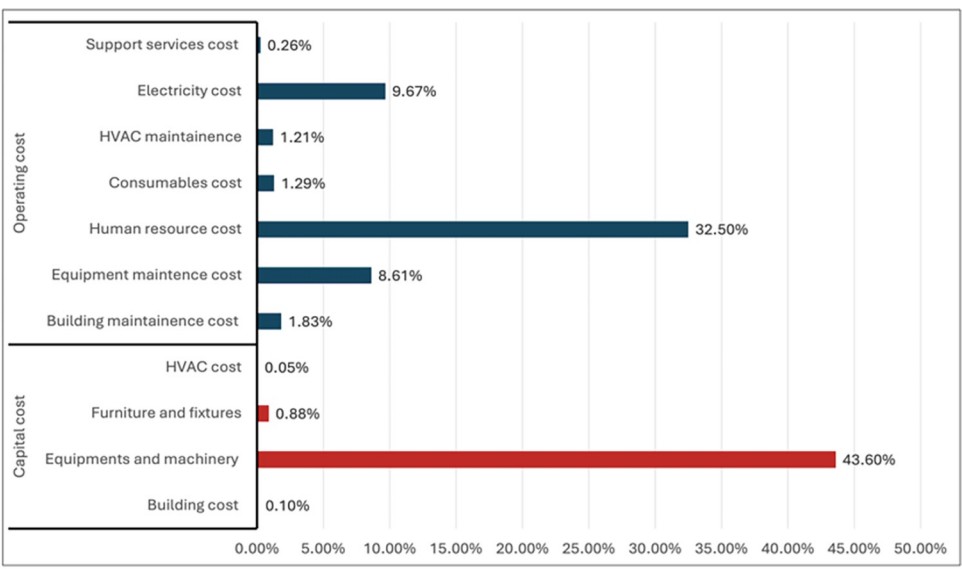

**Fig 2. Various cost centres of gamma knife facility at neurosurgery centre, AIIMS New Delhi.**

**Table 4. Total capital cost & operating cost required for Model A gamma knife facility having inhouse MRI facility.**

| S. No. | Capital cost | Operating Cost (Annual) |
|---|---|---|
| 1 | Building Cost*:<br>US $214,806 (INR 1,51,22,313) | Building Maintenance Cost:<br>US $ 29,364 (INR 20,67,253) |
| 2 | Machinery & Equipment Cost:<br>US $ 5,978,428 (INR 42,08,81,328) | Equipment Maintenance Cost:<br>US $138,478 (INR 97,48,857) |
| 3 | For Furniture & Fixtures:<br>US $ 67,483 (INR 47,50,768) | Human Resource Cost:<br>US $ 345,001 (INR 2,42,88,060) |
| 4 | HVAC Installation Cost:<br>US $ 12,784 (INR 9,00,000) | HVAC maintenance and operation Cost: US $ 19,432 (INR 13,68,000) |
| 5 | | Consumables Cost:<br>US $ 20,752 (INR 14,60,945) |
| 6 | | Electricity cost:<br>US $ 15,543 (INR 1,09,42,323) |
| 7 | | Manifold Cost:<br>US $ 2149 (INR 1,51,299) |
| 8 | | Laundry Cost:<br>US $ 284 (INR 20,006) |
| 9 | | BMW management Cost:<br>US $ 1693 (INR 1,19,208) |
| | Total Capital Cost:<br>US $ 6,273,500<br>(INR 44,16,54,409) | Total Operating Cost:<br>US $ 712,585 (INR 5,01,65,951)<br>Total Operating cost for 5 years:<br>US $ 3,562,923 (INR 25,08,29,755) |
| | Total Cost including operational cost for five years: US $ 9,836,423 (INR 69,24,84,164) | |

* Land cost was not taken into consideration

**Table 5. Total capital cost & operating cost required for Model B: Centre is only equipped with gamma knife & MRI scans are done from outside.**

| S. No. | Capital cost | Operating Cost (Annual) |
|--------|-------------|------------------------|
| 1 | Building Cost*:<br>US $ 169,073 (INR 1,19,02,724) | Building Maintenance Cost:<br>US $ 23,113 (INR 16,27,128) |
| 2 | Machinery & Equipment Cost:<br>US $ 4,728,865 (INR 33,29,12,102) | Equipment maintenance Cost:<br>US $ 98,251 (INR 69,16,857) |
| 3 | For Furniture & Fixtures:<br>US $ 55,450 (INR 39,02,947) | Human Resource Cost:<br>US $ 233,019 (INR 1,64,04,528) |
| 4 | HVAC Installation Cost:<br>US $ 12,784 (INR 9,00,000) | HVAC maintenance and operation Cost:<br>US $ 19,432 (INR 13,68,000) |
| 5 | | Consumables Cost:<br>US $ 20,752 (INR 14,60,945) |
| 6 | | Electricity Cost:<br>US $ 67,788 (INR 47,72,305) |
| 7 | | Manifold Cost:<br>US $ 1433 (INR 1,00,866) |
| 8 | | Laundry Cost:<br>US$ 284 (INR 20,006) |
| 9 | | BMW management cost:<br>US $ 1693 (INR 1,19,208) |
| | Total Capital Cost:<br>US $ 4,966,162 (INR 34,96,17,773) | Total Operating Cost:<br>US $ 465,765 (INR 3,27,89,843)<br>Total Operating cost for 5 years:<br>US $ 2,328,824 (INR 16,39,49,215) |
| | Total Cost including operational cost for five years: US $ 7,294,986 (INR 51,35,66,988) | |

* Land cost was not taken into consideration

Model A incurred significantly greater expenses, with an absolute capital cost of US $ 6,273,500 (INR 44,16,54,409) and operational costs amounting to US $ 3,562,923 (INR 25,08,29,755) in comparison to Model B with the capital cost of US $ 4,966,162 (INR 34,96,17,773) and operational costs of US $ 2,328,824 (INR 16,39,49,215).

## Discussion

Lower- and middle-income countries (LMICs) are at a disadvantage owing to a highly distorted ratio of the patient load and given the number as well as inequitable distribution of neurosurgeons. Additionally, scarcity of resources and the unstable economic condition of a vast majority of the population add on to the severity of condition. GKRS has evolved as a potent treatment for a number of neurosurgical entities, however, its availability is largely limited to

**Table 6. Break-even analysis for gamma knife facility in various scenarios.**

| Scenario | Working hours | Annual Procedures | Procedure charges (INR) | Variable cost | Break-even point |
|----------|---------------|-------------------|------------------------|---------------|------------------|
| **Scenario 1:** the number of procedures carried out in the Gamma Knife Facility each year remains fairly constant (651 procedures/annum) and the working hours remain the same. | 12 | 651 | US $ 2469 (INR 173831) | US $1367 (INR 96,239) | 3019 procedures (nearly 5 years) |
| **Scenario 2:** If the Gamma Knife Facility functions for 18 hours a day, which is the maximum permissible time limit for the MRI machine, the Fixed Cost remains the same. Also the relevant cost centres of the operating cost were also assumed to be increase by 50%. However, the overall variable cost per procedure comes down to 86956 assuming the number of procedures carried out also increases to 1.5 times considering there is a waiting time of up to 1 month at present. | 18 | 977 | US $1970 (INR 138658) | US$ 1235 (INR 86,956) | 4567 procedures (nearly 5 years) |
| **Scenario 3:** In case the facility charges the lowest prevalent rates per procedure in Delhi-NCR, which is Rs. 2,80,000 | 12 | 651 | US $ 3977 (INR 2,80,000) | US $1367 (INR 96,239) | 2749 procedures (nearly 4.2 years) |

the developed world and a few developing nations [15]. Most of the socioeconomic impact associated with open surgery for benign cranial base tumour stems from the indirect costs, such as loss of workdays and mortality. GKRS is a worthwhile treatment as it offers significant benefits to patients as well as to the society by reducing hospital stay and workdays lost in addition to reducing complications, mortality, socioeconomic loss, thereby improving cost-effectiveness.

According to a study conducted in Taiwan, the socioeconomic cost (US $34,453 +/- US $97,277) is higher for open surgery than for GKRS [(US $10,044 +/- US $7481), p < 0.01]. The Cost Effective Analysis (CEA) is significantly higher in gamma knife radiosurgery (US $3762 /quality-adjusted life year) than in open surgery [(US $8996 /quality-adjusted life year), p < 0.01] [16]. Gamma Knife may be cost competitive only if demand for SRS services is high enough to fully utilise the equipment working time. However, given low patient demand and competing radiotherapy needs, GKRS is apparently more costly [17]. Once a unit is treating more than 100 patients per year the Gamma Knife becomes the most cost-effective technology by a factor of almost 100% [18].

Major cost drivers in a Gamma Knife Facility were machinery & equipment, manpower, electricity and equipment maintenance cost. Highest capital cost was incurred on Machinery & Equipment, which is similar to what has been observed in facilities having sophisticated high end technology [19]. In a study on cost analysis of treating cardiovascular diseases, it was observed that the highest cost was incurred on equipment followed by human resource [20]. Similar findings have also been reported in another study done in a public hospital to assess the cost of operation theatre services [21].

A study conducted on patients of vestibular schwannomas, meningiomas, brain metastases, and arteriovenous malformations and concluded that the average overall costs for primary microsurgical therapy were €10,814 ± €6,108 out of which €1,417 ± €426 was particularly attributed to the surgical procedure [22]. This approximately amounts to US $1790 (INR 1,25,000±37,000) which is comparable to the cost per procedure at the institute US $2469 (INR 1,73,832) [13]. Zygourakis et al observed that the total cost for patients who underwent Gamma knife surgery for vestibular schwannomas was $9,737 ± $5,522 (approximately INR 7,00,000 ± 4,40,000) [23].

Breakeven analysis is essentially needed for any investment for deciding upon the user charges especially in privately funded healthcare facilities so as to ensure the financial sustainability and giving an insight into the extent of utilisation/units needed for any healthcare service being provided. It is much needed for evidence based decision making especially in resource constrained settings of Lower and Middle Income Countries (LMIC) where there are essential competing requirements. In this study, it was observed that user charges being levied for GKRS are lower than the operational charges, which makes it impossible for achieving breakeven, however, being one of the apex research and teaching publicly funded institute with a mission to provide a world class treatment at an affordable rates, the cost to the patient has been kept highly subsidised. If compared against the user charges levied by private sector hospitals for gamma knife procedure, the existing user charges at the institute would be less than 1/4th of the charges levied by the private hospitals.

In addition, given the healthcare reforms in the country under the pioneering Pradhan Mantri Ayushman Bharat Yojana (PM-JAY), findings of this study becomes pertinent which can aid policy makers in arriving at estimates/packages for financing of the healthcare delivery for GKRS. Ayushman Bharat, a flagship scheme of Government of India, was launched as recommended by the National Health Policy 2017, to achieve the vision of Universal Health Coverage (UHC). This initiative has been designed to meet Sustainable Development Goals (SDGs) and its underlining commitment, which is to "leave no one behind".

This study has been conducted in a leading Neurosciences Centre of the Country with huge patient load which leads to increased waiting times as the treatment is subsidised, with diverse procedures being carried out which gives a variable case load for cost analysis making it comprehensive and inclusive. The cost analysis of Gamma Knife Radio Surgery is done as it will facilitate better value of the services as well as benchmark for replicating developmental activities in the future in other parts of the country as well as for reviewing the policy guidelines on budgeting of Heath Care facilities. An attempt has been made to include the major cost centres except the cost of land which might have led to underestimation of the unit cost and cost estimation for establishing gamma knife facility. Various cost heads like Administrative Costs, Computer Facility, Medical Records, IT costs etc. would have been insignificant cost heads in comparison to the overall cost and its identification would have taken significant resources, hence, have been excluded from the study.

Hospital/Healthcare Managers and Clinicians should strive for possible reduction of costs through linking it to a greater optimization of the use of the machine, to bring down the pricing of such high-cost treatment modality for the public welfare. Therefore, identifying patients with clear indications for this type of treatment, periodic maintenance checks, specifically trained personnel etc. are the actions that would lead to the optimization of the use of the GKRS with possible reduction of costs. This has a direct implication on the availability, accessibility and affordability of high-cost treatment modality with specific relevance to resource constrained countries. Availability of data posed a significant challenge given the existing practices of record management pertaining to resource consumption. Better record keeping practices especially focussing on the resource utilisation for healthcare services is sine qua non for cost analysis, which is often a limitation in public healthcare delivery system in the country. It will also lead to better appreciation of the quantum of resource utilisation by the healthcare managers nudging them to ensure the optimal utilisation of finite healthcare resources.

## Conclusion

It is costly to establish and operate a Gamma Knife Facility, therefore incorporating both cost and break-even analysis for healthcare services is essential for optimal utilisation of resources and financial sustainability. Policymakers need to be aware of these financial assessments to standardize practices, create or update relevant healthcare packages, and ensure that the system remains aligned with the technological advancements. This strategic approach will facilitate the sustainable integration of high-end medical technologies into healthcare delivery, thus serving as a crucial reference for decision-makers seeking to strike a balance between affordability and financial sustainability in delivering cutting-edge medical treatments within a public sector framework for achieving the overall goal of healthcare for all.

## Acknowledgments

We sincerely acknowledge and appreciate the contribution of various healthcare workers of gamma knife facility for helping in data acquisition for arriving at cost analysis. Support provided by various sections and divisions such as nursing establishment, hospital stores, engineering services, accounts department of AIIMS New Delhi is deeply appreciated.

## Author Contributions

**Conceptualization:** Nitin Agrawal, Nishant Sharma, Vijaydeep Siddharth, Sidhartha Satpathy.

**Data curation:** Nitin Agrawal, Kshitija Singh.

**Formal analysis:** Nitin Agrawal, Nishant Sharma, Kshitija Singh.

**Methodology:** Nitin Agrawal, Nishant Sharma, Kshitija Singh, Vijaydeep Siddharth, Sidhartha Satpathy.

**Project administration:** Nitin Agrawal, Vijaydeep Siddharth.

**Resources:** Nitin Agrawal.

**Supervision:** Nishant Sharma, Vijaydeep Siddharth, Sidhartha Satpathy.

**Validation:** Nitin Agrawal, Nishant Sharma, Tilotma Jamwal, Vijaydeep Siddharth.

**Visualization:** Tilotma Jamwal, Vijaydeep Siddharth.

**Writing – original draft:** Nitin Agrawal, Nishant Sharma, Tilotma Jamwal, Vijaydeep Siddharth.

**Writing – review & editing:** Nitin Agrawal, Nishant Sharma, Tilotma Jamwal, Vijaydeep Siddharth.

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
