## [Decision Letter · Decision Letter 0]

19 Nov 2024

PONE-D-24-23739Optimizing costs and sustainability for Gamma Knife Radiosurgery: A cost and breakeven analysis at India's largest Neurosurgery CentrePLOS ONE

Dear Dr. Siddharth,

Thank you for submitting your manuscript to PLOS ONE. After careful consideration, we feel that it has merit but does not fully meet PLOS ONE’s publication criteria as it currently stands. Therefore, we invite you to submit a revised version of the manuscript that addresses the points raised during the review process.

We look forward to receiving your revised manuscript.

Kind regards,

Mohammed Misbah Ul Haq, Pharm-D

Academic Editor

PLOS ONE

Journal Requirements: When submitting your revision, we need you to address these additional requirements. 1. Please ensure that your manuscript meets PLOS ONE's style requirements, including those for file naming. The PLOS ONE style templates can be found at https://journals.plos.org/plosone/s/file?id=wjVg/PLOSOne_formatting_sample_main_body.pdf and https://journals.plos.org/plosone/s/file?id=ba62/PLOSOne_formatting_sample_title_authors_affiliations.pdf 2. We note that your Data Availability Statement is currently as follows: All relevant data are within the manuscript and its Supporting Information files. Please confirm at this time whether or not your submission contains all raw data required to replicate the results of your study. Authors must share the “minimal data set” for their submission. PLOS defines the minimal data set to consist of the data required to replicate all study findings reported in the article, as well as related metadata and methods (https://journals.plos.org/plosone/s/data-availability#loc-minimal-data-set-definition). For example, authors should submit the following data: - The values behind the means, standard deviations and other measures reported;- The values used to build graphs;- The points extracted from images for analysis. Authors do not need to submit their entire data set if only a portion of the data was used in the reported study. If your submission does not contain these data, please either upload them as Supporting Information files or deposit them to a stable, public repository and provide us with the relevant URLs, DOIs, or accession numbers. For a list of recommended repositories, please see https://journals.plos.org/plosone/s/recommended-repositories. If there are ethical or legal restrictions on sharing a de-identified data set, please explain them in detail (e.g., data contain potentially sensitive information, data are owned by a third-party organization, etc.) and who has imposed them (e.g., an ethics committee). Please also provide contact information for a data access committee, ethics committee, or other institutional body to which data requests may be sent. If data are owned by a third party, please indicate how others may request data access. 3. Please ensure that you refer to Figure 1 in your text as, if accepted, production will need this reference to link the reader to the figure. 4. Please review your reference list to ensure that it is complete and correct. If you have cited papers that have been retracted, please include the rationale for doing so in the manuscript text, or remove these references and replace them with relevant current references. Any changes to the reference list should be mentioned in the rebuttal letter that accompanies your revised manuscript. If you need to cite a retracted article, indicate the article’s retracted status in the References list and also include a citation and full reference for the retraction notice.

**Additional Editor Comments:**

Your manuscript, "Optimizing costs and sustainability for Gamma Knife Radiosurgery: A cost and breakeven analysis at India's largest Neurosurgery Centre," addresses a critical and underexplored topic in healthcare economics. The study provides valuable insights into cost optimization and sustainability in the context of advanced neurosurgical care.

To enhance the manuscript further, consider elaborating on the methodology, particularly the classification of fixed and variable costs, to improve clarity and reproducibility. Expanding on the potential implications for patient access, clinical outcomes, and healthcare policy would strengthen the discussion. Additionally, comparisons with similar analyses from other regions or international contexts could provide a broader perspective and highlight the global relevance of your findings. Overall, your study is well-conceived and offers significant potential for guiding cost-effective practices in Gamma Knife Radiosurgery.

Reviewers' comments:

Reviewer's Responses to Questions

**Comments to the Author**

1. Is the manuscript technically sound, and do the data support the conclusions?

Reviewer #1: Partly

Reviewer #2: Yes

2. Has the statistical analysis been performed appropriately and rigorously? 

Reviewer #1: Yes

Reviewer #2: N/A

3. Have the authors made all data underlying the findings in their manuscript fully available?

Reviewer #1: No

Reviewer #2: Yes

4. Is the manuscript presented in an intelligible fashion and written in standard English?

Reviewer #1: Yes

Reviewer #2: Yes

5. Review Comments to the Author

Reviewer #1: Optimizing costs and sustainability for Gamma Knife Radiosurgery: A cost and breakeven analysis at India's largest Neurosurgery Centre

The study in the manuscript is important, as there is a scarcity of primary healthcare costing data in India. Authors claim to provide cost of GKRS procedure in the Indian contest. The context was well set in the introduction. I appreciate all the authors who have contributed to generating this manuscript. The manuscript show sufficient potential that the authors should be encouraged to resubmit a revised version. I have some comments on the manuscript below.

1. Mention the software used for calculation. (I suppose all calculations were done in Microsoft Excel. Mention is along with the version of software)

2. Nominal interest rate was not mentioned. Is it equivalent to the discount factor generally used in costing studies?

3. The author should provide a number of Human resources with their time contribution to GKRS. (May be in supplemental material)

4. Is there any overhead charges for the administration department (which supports the overall organization where the facility is established) included in the cost analysis? If not please give reason (it may be included in limitation)

5. Why 5 years cost was calculated? Where you have used it in analysis?

6. In break-even analysis, the fixed cost = US$ 717,509 (INR5,05,12,665) (Capital Cost) is kept constant as numerator. Is it annual? Or per procedure?

And in the denominator, procedure charge – variable cost (as per the equation on page 10, line 177) = 115000-96239 = 18761. So the breakeven point must be = 5,05,12,665/18761 = 2692, which is not equal to the break-even point mentioned in scenario 1 in Table 6. Can you explain how do you calculated it?

7. I recommend that the author provide a raw data file with all the data required to reproduce the values mentioned in the manuscript as supplemental material.

8. Label the X axis for figure 2.

9. Figure legends were not given

10. In figure 3, please mention whre actually the break-even point is (crossing the blue and red line, I suppose)

11. In Figure 3 C, the Y-axis shows one “0” extra. Is this a mistake, or did you get the results you expected?

12. The values mentioned in Figure 3 a,b titles are not in the table or text? Please explain who you derive this value.

13. Figure 3 quality is poor. (May be you can revise fonts and pixel size)

Reviewer #2: I read with interest the study by Agrawal et al. The authors, in this study, analyze the costs related to the use of gamma knife radio surgery in the largest neurosurgery center of an Institute of National Importance, in Delhi, India.

The reported data clearly reflect the legislative regulations adopted in India. The costs highlighted appear to be important, both those related to the operation of the equipment (building, water, refrigeration, energy) and those related to the dedicated personnel (healthcare personnel, technicians). The possible reduction of costs should be linked to a greater optimization of the use of the machine. Therefore, identifying patients with clear indications for this type of treatment, periodic maintenance checks and specifically trained personnel are actions that would lead to the optimization of the use of the treatment with possible reduction of costs.

The comparison reported in the text between patients treated with GKRS and patients treated surgically does not seem pertinent to me. In patients undergoing surgical treatment there are numerous variants that must be considered (prolonged hospitalization, pharmacological therapy, complications). In my opinion this part could be removed from the text.

6. PLOS authors have the option to publish the peer review history of their article (what does this mean?). If published, this will include your full peer review and any attached files.

Reviewer #1: No

Reviewer #2: **Yes: **Gerardo Caruso

---

## [Author Response · Author response to Decision Letter 0]

9 Dec 2024

Response to Editors, Additional editor and reviewers have been made pointeise and incorporated changes whereever needed.

---

## [Decision Letter · Decision Letter 1]

15 Dec 2024

PONE-D-24-23739R1Optimizing costs and sustainability for Gamma Knife Radiosurgery: A cost and breakeven analysis at India's largest Neurosurgery CentrePLOS ONE

Dear Dr. Siddharth,

Thank you for submitting your manuscript to PLOS ONE. After careful consideration, we feel that it has merit but does not fully meet PLOS ONE’s publication criteria as it currently stands. Therefore, we invite you to submit a revised version of the manuscript that addresses the points raised during the review process.

We look forward to receiving your revised manuscript.

Kind regards,

Mohammed Misbah Ul Haq, Pharm-D

Academic Editor

PLOS ONE

Journal Requirements:

Reviewers' comments:

Reviewer's Responses to Questions

**Comments to the Author**

1. If the authors have adequately addressed your comments raised in a previous round of review and you feel that this manuscript is now acceptable for publication, you may indicate that here to bypass the “Comments to the Author” section, enter your conflict of interest statement in the “Confidential to Editor” section, and submit your "Accept" recommendation.

Reviewer #2: All comments have been addressed

2. Is the manuscript technically sound, and do the data support the conclusions?

Reviewer #2: Yes

3. Has the statistical analysis been performed appropriately and rigorously? 

Reviewer #2: Yes

4. Have the authors made all data underlying the findings in their manuscript fully available?

Reviewer #2: Yes

5. Is the manuscript presented in an intelligible fashion and written in standard English?

Reviewer #2: Yes

6. Review Comments to the Author

Reviewer #2: (No Response)

7. PLOS authors have the option to publish the peer review history of their article (what does this mean?). If published, this will include your full peer review and any attached files.

Reviewer #2: No

---

## [Author Response · Author response to Decision Letter 1]

16 Dec 2024

List of referenes have been reviewed thoroughly with minor changes.

No retracted article found. 

No changes to the list of references

---

## [Editor Report · Decision Letter 2]

18 Dec 2024

Optimizing costs and sustainability for Gamma Knife Radiosurgery: A cost and breakeven analysis at India's largest Neurosurgery Centre

PONE-D-24-23739R2

Dear Dr. Siddharth,

We’re pleased to inform you that your manuscript has been judged scientifically suitable for publication and will be formally accepted for publication once it meets all outstanding technical requirements.

Kind regards,

Mohammed Misbah Ul Haq, Pharm-D

Academic Editor

PLOS ONE
---

## [Editor Report · Acceptance letter]

27 Dec 2024

PONE-D-24-23739R2 

PLOS ONE

Dear Dr. Siddharth, 

I'm pleased to inform you that your manuscript has been deemed suitable for publication in PLOS ONE. Congratulations! Your manuscript is now being handed over to our production team.

Kind regards, 

on behalf of

Dr. Mohammed Misbah Ul Haq 

Academic Editor

PLOS ONE